# Vibration Event Recognition Using SST-Based Φ-OTDR System

**DOI:** 10.3390/s23218773

**Published:** 2023-10-27

**Authors:** Ruixu Yao, Jun Li, Jiarui Zhang, Yinshang Wei

**Affiliations:** 1School of Safety Science and Engineering, Xi’an University of Science and Technology, Xi’an 710054, China; 18729228435@163.com (R.Y.); jiaruiziy@163.com (J.Z.);; 2Shaanxi Provincial Key Laboratory of Coal Fire Disaster Prevention, Xi’an 710054, China

**Keywords:** vibration signal, distributed fiber vibration, classification

## Abstract

We propose a method based on Synchrosqueezing Transform (SST) for vibration event analysis and identification in Phase Sensitive Optical Time-Domain Reflectometry (Φ-OTDR) systems. SST has high time-frequency resolution and phase information, which can distinguish and enhance different vibration events. We use six tap events with different intensities and six other events as experimental data and test the effect of attenuation. We use Visual Geometry Group (VGG), Vision Transformer (ViT), and Residual Network (ResNet) as deep classifiers for the SST transformed data. The results show that our method outperforms the methods based on Continuous Wavelet Transform (CWT) and Short-Time Fourier Transform (STFT), while ResNet is the best classifier. Our method can achieve high recognition rate under different signal strengths, event types, and attenuation levels, which shows its value for Φ-OTDR system.

## 1. Introduction

Distributed fiber optic sensing system is a technology that uses optical fiber as a sensor, which can realize real-time monitoring of temperature [1], vibration, sound, and other physical quantities along the optical fiber. A distributed fiber optic sensing system has the advantages of wide coverage, high sensitivity, strong anti-interference ability, etc., and is widely used in oil and gas pipelines, bridges, tunnels, seismology, and other fields [2]. However, distributed fiber optic sensing systems also face some challenges, one of which is how to effectively process and classify the signals collected from optical fibers to identify different types of events and anomalies, which has become a focal point of concern within the field of distributed fiber optic intrusion detection.

To solve this problem, many researchers have proposed different signal processing and classification methods. Time-domain [3] methods obtain a single piece of information and, therefore, fewer types can be accurately identified. Frequency-domain [4] feature analysis methods can obtain the intrinsic spectral characteristics of the signal, but they may ignore the time-domain information for non-stationary signals that are constantly changing over time. Therefore, the time-frequency domain characterization method is the most commonly used approach, which transforms the signal at different times and frequencies to extract the local features of the signal.

Many methods based on time-frequency domain features have been proposed, such as wavelet decomposition, STFT, etc. In 2015 [5], Wu et al. used wavelet decomposition to extract the spectral distribution as a feature vector and combined it with Back Propagation (BP) neural network to classify the events of environmental noise, system drift, and man-made intrusion, with an accuracy of 89.19%. In 2018 [6], Xu C et al. used the spectral subtraction method and STFT to get the spectral map, and then inputted it into a Convolutional Neural Network (CNN) for automatic feature extraction and classification. The recognition rate of the four events of digging, walking, vehicle overtaking, and sabotage exceeded 90% in each case. In 2022 [7], Xu W et al. calculated the difference between two adjacent Rayleigh Back Scattering (RBS) traces and normalized the difference results of the images. The images were labeled, tagged, and then recognized by You Only Live Once (YOLO) as calm state, rigid collision with the ground, impact protection network, vibration protection network, and cutting protection network of five states and localization; a recognition rate of 96.14% was attained. In 2023 [8], Du X et al. performed wavelet transform and feature extraction on the original signal, and then optimized the feature vector using the sample feature correction algorithm. Support Vector Machines (SVM)was then used to recognize watering, climbing, knocking, pressure, as well as a spurious perturbation event. SVM achieved a 98.5% recognition rate, as well as a 99.8% recognition rate for spurious perturbation events. All of these methods, to varying degrees, quickly localized intrusion events by classifying several explicit events, which is a significant improvement for the security of the monitoring system. However, these methods also have some limitations, such as restricted time-frequency resolution and ignored phase information.

These limitations are overcome by a novel method for signal processing and classification based on SST that is presented in this paper. CWT [9] is a time-frequency analysis tool that transforms signals at different scales and translations to extract local features of the signals. SST [10] is a technique that takes advantage of the sparsity of the signals to reconstruct the signals through a small number of measurements. In this paper, we perform signal extraction and reconstruction of vibration signals using SST and use the phase information of CWT to enhance the feature representation of the signal. A dataset of vibration signals generated in the laboratory is used, which contains 12 categories of signals generated by different strengths of tap and different events. We also consider the effect of signal attenuation on the classification performance and process and classify the attenuated weak signals. We compare the classification effect of three kinds of time-frequency maps, namely STFT, CWT, and SST, in the VGG [11], ViT [12], and ResNet [13] algorithms. The experimental results show that our method can stably and effectively improve the classification performance and provide a new way for the application of the Φ-OTDR system.

## 2. Systems and Algorithms

### 2.1. *Φ*-OTDR System

The Φ-OTDR system was originally developed from the OTDR system [14], in which the phase of the Rayleigh backscattered signal from the fiber optic cable depends only on the distance. However, when a highly coherent light source is used, the backscattered coherent light within the optical pulse will interfere and lead to an intensity distribution related to the phase difference of the interfering light signal. A Φ-OTDR system is a new distributed fiber optic acoustic sensing technology which is based on the principle of acoustic phase sensing, and, through spatial differential interference technology, it demodulates the phase information of backward Rayleigh-scattered light, which can then realize the reconstruction of external vibration signals. When an external intrusion event acts on the sensing fiber, the length L of the fiber in the sensing area and the refractive index n of the fiber core will change, which will lead to the change of the phase of the light wave in the fiber. By calculating the difference of backward Rayleigh scattered light at different moments, the location of the intrusion event can be accurately located. The wave function of the intrusion signal can be obtained by the Φ-OTDR system, and then the amplitude, frequency, phase, and other information of the intrusion signal can be obtained by the demodulation of the host computer. The Φ-OTDR system has a high signal-to-noise ratio. For intrusion signal pattern recognition, complete and accurate extraction of the original signal is an important prerequisite for feature extraction. The better the signal restoration, the more accurate the extracted features as the basis for judging the intrusion event, which can then be accurately classified and recognized.

To demonstrate the signal processing and classification of the Φ-OTDR system, a laboratory setup was built as described in this paper. Figure 1 shows the schematic diagram of the setup. The device uses a narrow linewidth laser (NLL) with BG-PL-1550-20-3K technology, which has a linewidth of 3 kHz and produces high-quality pulse light. The NLL operates at a wavelength of 1550.12 nm and a power of 20 mW, with a pulse width of 100 ns and a repetition rate of 1 kHz. The light from the NLL passes through an isolator (IO), which has an isolation degree of 40 dB and prevents the back-reflected light in the fiber from affecting the laser. The IO is an IO-FS50 model from Thorlabs. After passing through an acousto-optic modulator (AOM), the light is amplified by an erbium-doped fiber amplifier (EDFA) to enhance the sensitivity and dynamic range of the system. The AOM is a CETC44-AOM-200M-5V-1W model from China Electronics Technology Group 44 Institute, with a working frequency of 200 MHz. The EDFA is an AEDFA-13-M model from Poffer Optoelectronics Technology, with a gain of 15 dB. The photodetector (PIN) receives the light signals reflected or scattered from the sensing fiber and converts them into electrical signals. The PIN is a PDA10CS model from Thorlabs, with a response wavelength range of 800 nm to 1700 nm. The data acquisition card (DAQ) collects and processes the signal after photoelectric conversion, using a PCle-9802DC model customized by Jianyi Technology, with a maximum sampling rate of 250 MSa/s. The sensing fiber and buried cable are Corning SMF-28 single-mode fibers, with a core optical refractive index of 1.4.

The purpose of this paper is to introduce experiments that are conducted in a laboratory environment to collect and analyze different types of signals applied onto the sensing optical fiber cable. The cable has a length of 11 km and was divided into two sections: the front section and the rear section. The front section was wrapped around wheels and left exposed to the air, while the rear section was buried 30 cm deep in soil. The first task of the experiment was to detect a phone vibration (PV) signal within the front section of the optical fiber. This was achieved by placing a mobile phone on the cable to create vibrations, and then detecting the signal from the beginning of the fiber optic cable over a distance of 2–4 km. For the second task, the rear section of the optical fiber was used to detect eleven other signals. These signals were generated by various activities performed on or within the soil, such as tapping, stamping, chiseling, mixing, watering, and digging. Sampling points from 9.94–10 km were used for this purpose. The tapping signal varied in intensity based on factors such as force, frequency, and even the gender of the individual performing the action. Taps performed by females were categorized as low force, while those performed by males were considered strong force. The time interval between each tap ranged from 1 s to 3 s, representing fast, normal, and slow tapping speeds, respectively. The stamping (ST) signal was generated by an individual forcefully stepping on the soil. The chiseling (CH) signal was created by inserting a chisel into the soil. The excavation walking and tapping (MI) signal resulted from three simultaneous actions on the soil: tapping, stamping, and chiseling. The watering (WA) signal was produced by pouring water into the soil. Lastly, the digging (HSZ) signal was generated using a Hyrule shovel to dig into the soil. Various devices and instruments were utilized throughout this experiment to generate, transmit, receive, and process these signals within the optical fiber cable. A schematic diagram of this experimental setup can be seen in Figure 1.

### 2.2. SST Principle

To realize signal processing and classification of the Φ-OTDR system, we will introduce a method based on SST. SST is a technique that leverages the sparsity of the signals to reconstruct them from a small number of measurements.

Daubechies et al., one of the founders of wavelets, proposed SST as an empirical modal decomposition tool in 2011 [10]. SST rearranges the time-frequency coefficients through the synchronous compression operator, which shifts the time-frequency distribution of the signal at any point of the time-frequency plane to the center of gravity of the energy and enhances the energy concentration of the instantaneous frequency. Therefore, the core of SST is the estimation of the instantaneous frequency. It can better solve the time-frequency ambiguity problem existing in the traditional time-frequency analysis method. As a special rearrangement, SST can not only sharpen the time-frequency representation, but also recover the signal.

Generally speaking, the Morlet wavelet function is more suitable for analyzing periodic or quasi-periodic signals, while the generalized Morse wavelet (GMW) function is more suitable for analyzing signals with sharp changes or sudden events. Therefore, in this paper, we used the GMW wavelet function. We filled the signal by reflection filling, which can make its length to the nearest power of 2. We set the threshold of the control synchronization squeezing operator to 0.05.

As shown in Equation (1), SST improves the time-frequency resolution by compressively rearranging the CWT transform coefficients in the frequency direction and rewriting $\Psi f\left(a,b\right)$ as the CWT of the signal f in the frequency domain according to Plancherel’s principle:(1)$$\Psi f\left(a,b\right)=\frac{1}{2\pi }\int_{-\infty }^{+\infty }\hat{f}\left(\xi \right)a^{1/2}\bar{\hat{\psi }\left(a\xi \right)}e^{ib\xi }d\xi$$
where ξ is the angular frequency and $\hat{f}\left(\xi \right)$ is the Fourier transform of the signal f. When the wavelet transform coefficients at any point (a,b) are not equal to 0, the instantaneous frequency of the signal f is $\psi f\left(a,b\right)$, as shown in Equation (2):(2)$$\psi f\left(a,b\right)=-i{\left(\Psi f(a,b)\right)}^{-1}\frac{\partial }{\partial b}\Psi f(a,b)$$

Then a synchronized compression transformation is performed to create a mapping from the starting point (b, a) to (b, Ψf(a,b)), and Ψf(a,b), etc. are transformed from the time-scale plane to the time-frequency scale plane to obtain a new time-frequency map.

Then the frequency variable ψ, the scale variable a, and the translation factor b are discretized, and Ψf(a,b) can be obtained when it satisfies $a_{k}-a_{k-1}={\left(\Delta a\right)}_{k}$ only at the point $a_{k}$, and the corresponding $Tf\left(a,b\right)$ of the SST is similarly computed when it satisfies $\psi_{l}-\psi_{l-1}=\Delta \psi$ in the continuous interval [$\psi -\frac{1}{2}\Delta \psi ,\;\psi +\frac{1}{2}\Delta \psi$].

The final expression of SST is shown in Equation (3):(3)$$Tf\left(\psi_{l},b\right)=\Delta \psi^{-1}\sum_{a_{k:\left|\psi \left(a_{k,b}\right)-\psi_{l}\right|}}\Psi f\left(a_{k},b\right){a_{k}}^{-\frac{3}{2}}{\left(\Delta a\right)}_{k}$$

### 2.3. Analog Signal Verification

Generally, the frequencies of vibration events are concentrated in the low frequency range [15]. Therefore, we randomly generated simulated signals at three frequencies within 0–100 Hz, and performed time-frequency analysis of the simulated signals using CWT and SST. We compared the time-frequency resolution and phase information of the two methods. From Figure 2, we can see that signals with different amplitudes at different frequencies have different characteristics in the time-frequency domain, and SST has a higher time-frequency resolution and clearer phase information than CWT.

### 2.4. Recognition Algorithm Flow

As shown in Figure 3, a novel method for signal processing and classification based on SST and deep network is presented in this paper. The algorithm consists of the following steps:(1)The original denoised vibration signal data are obtained and CWT is performed on the signal to obtain its wavelet transform coefficients in the frequency domain. CWT is a tool for time-frequency analysis that can transform the signal at different scales as well as translations, so as to extract its local features.(2)The instantaneous frequency of the wavelet transform coefficients is calculated when they are not zero, and SST is performed on the instantaneous frequency to obtain the SST time-frequency map. SST is an improved CWT method, which preserves the phase information of the signal and removes the interference of the cross-terms, thus improving the time-frequency resolution.(3)STFT is performed on the signal to obtain the STFT time-frequency map. STFT is a commonly used tool for time-frequency analysis, which can Fourier transform the signal under different time windows to obtain its local spectrum.(4)The time-frequency maps are categorized and organized into samples, which are classified and recognized by deep networks. In this paper, three deep networks, namely VGG, ViT, and ResNet, are used as classifiers to recognize the SST-transformed data. Deep network is a machine learning model based on multi-layer neural network, which can realize automatic feature extraction and classification of image data.

## 3. Experimental Analysis

### 3.1. Dataset

In order to verify the classification performance of our method in complex situations, we collected data of different strengths and types of vibration events in the laboratory as the actual vibration event dataset. Six types of tapping with different strengths were collected for this paper, namely tap slow force (TSF), tap fast force (TFF), tap normal force (TNF), tap slow little force (TSLF), tap fast little force (TFLF), and tap normal little force (TNLF). These tapping events were generated by applying taps of different strengths at different locations and in different directions on the sensing fiber to simulate the behavior of different intruders. In order to increase the complexity of the environment, we also collected data from six different types of events, including HSZ, WA, MI, ST, PV, CH. These events were generated by performing different activities near or above the sensing fiber to simulate different environmental disturbances. We uniformly cut the signals every 3 s according to the latest time of the complete occurrence of each event as a sample signal, and then we performed CWT, SST, and STFT on the sample signals to generate time-frequency maps. Finally, we categorized and organized the time-frequency map dataset. As shown in Table 1 and Table 2, we collected a total of 3840 sample signals, with 1920 sample signals for each category, including 320 sample signals for each type of percussive vibration event.

The original waveform vibration images of the 12 data are shown in Figure 4. It can be seen that there are obvious differences in the wave peaks, frequencies, and morphology of different data. Based on these differences, this paper uses time-frequency analysis to extract the features of the signals and classify and identify them. CH has three raised wave peaks with maximum intensity at 0.1, which is similar to the MI vibration image. Both of them experience a sudden increase in wave peaks, but the intensity of MI is greater than that of CH at around 0.4, and the number of wave peaks is higher. There are four, a phenomenon which indicates that both CH and MI are generated by intermittent events. HSZ exhibits one raised wave with a maximum intensity around 0.1, indicating that HSZ is generated by a smaller shock event. PV exhibits a continuously increasing wave with a maximum intensity of 0.24, ST displays a raised wave with a maximum intensity of 0.075, and WA displays a continuously decreasing wave with a maximum intensity of −0.025. These findings suggest that PV, ST, and WA are all generated by persistent events, but the intensity of the event responsible for PV increases gradually, the event intensity of ST is smaller and stable, and the event intensity of WA decreases gradually. TFF exhibits several consecutive wave crests with a maximum intensity of around 0.3, TNF exhibits four wave crests with a maximum intensity of 0.4, which is similar to that of MI although the wave crests decay and oscillate a little bit longer, and TSF exhibits two wave crests with a maximum intensity of 0.2. The number of wave crests raised in TSF is similar to that of CH, but the intensity of TSF is higher than that of CH at around 0.4. The wave crests of TFLF rise continuously with a maximum intensity of around 0.2, TNLF exhibits three wave crests with a maximum intensity of around 0.4, and TSLF exhibits two wave crests with a maximum intensity of 0.2. This suggests that TFF, TNF, TSF, TFLF, TNLF, and TSLF are all generated by tapping events with different strengths and velocities, of which TFF and TFLF are examples of fast tapping, TNF and TNLF are examples of normal tapping, and TSF and TSLF are examples of slow tapping.

As shown in Figure 5, the vibration images of the attenuated signals for 12 datasets illustrate the effect of the attenuation process on the signal strength and characteristics. The figures show that the signal amplitudes have slightly diminished compared to the original data, but the overall characteristics have not changed much. This suggests that the attenuation process preserves the quality of the data.

### 3.2. Experimental Results and Analysis

In this paper, events are recognized by a deep classification network and some commonly used evaluation metrics for image classification are used to measure the performance of the model on the image classification task. The evaluation metrics for deep network image classification are as follows:(1)Accuracy (Accuracy) refers to the ratio of the number of correctly classified samples to the total number of samples: the higher, the better. It is a global metric that can reflect the performance of the model on the whole dataset, but cannot reflect the difference of the model on different categories.(2)Confusion Matrix (Confusion Matrix) is used to show the correspondence between the model’s prediction results and the actual results for each category, from which indicators such as Precision, Recall, True Rate, False Positive Rate, etc. can be calculated. It is a type of localized index, which can reflect the performance of the model on each category, as well as the ability of the model to distinguish between different categories.(3)F-value (F1-score) is the reconciled average of the check accuracy rate and the check completeness rate, which integrally reflects the precision and recall of the model: the higher, the better. It is a balanced metric that can be used to compare the performance of different models on different datasets, and is especially applicable to the case of unbalanced datasets.

#### 3.2.1. Raw Data Identification

Figure 6 shows the time-frequency diagrams of CWT, SST, and STFT for 12 types of raw signals, and it can be seen that the time-frequency diagrams of each type of event have different features, which are analyzed in this paper, and the time-frequency diagrams were automatically feature-extracted and categorized using deep networks. From Figure 6, it can be observed that the performance of the three time-frequency analysis methods, namely CWT, SST and STFT, varies significantly depending on the events. The time-frequency diagrams of CWT have higher frequency resolution but lower time resolution, so they can clearly show the spectral distribution of the signal but cannot accurately show the temporal variations of the signal. The time-frequency diagrams of SST have higher temporal resolution and phase information but lower frequency resolution, so SST can clearly display the time variation and phase variation of the signal but cannot accurately display the spectral distribution of the signal. The time-frequency diagram of STFT has fixed time resolution and frequency resolution, so it can display the time variation and spectral distribution of the signal in a balanced way but cannot display the phase information of the signal.

As part of this study, events were recognized by deep classification network, and evaluation metrics such as accuracy, F1-score, and time were used to measure the classification performance of the model under different time-frequency analysis methods. As seen in Table 3 and Figure 7 for six signals with different intensities, the SST Resnet model has the highest recognition accuracy of 99.48%, which is the best performance, and the CWT Resnet model is the second best with 98.18% recognition accuracy. For the recognition accuracy of the six intensity signals under the six background signals, SST ResNet was the best performing model with 96.74% recognition accuracy, followed by CWT ResNet with 94.66% recognition accuracy. These results indicate that the SST ResNet model is able to effectively distinguish different intensities and types of percussive vibration events with high robustness and generalization ability.

Different time-frequency analysis methods and different deep network models have obvious differences in classification performance. Among the three time-frequency analysis methods, namely CWT, SST, and STFT, SST was found to attain the best classification, followed by CWT, with STFT attaining the worst results. This indicates that SST is the most suitable time-frequency analysis method because it can retain the phase information of the signal and eliminate the interference of the cross terms, which improves the time-frequency resolution and classification performance. Among the three deep network models, namely VGG, ViT, and ResNet, the classification performance of ResNet was found to outperform the others, followed by ViT, with VGG attaining the worst results. This indicates that ResNet is the most suitable deep network model because it enables fast localization and recognition of image data, has a deeper network structure and fewer parameters compared to VGG [16], and differs from ViT [17] in the way it handles representations, which has mixed local and global information at both the lower and higher levels. In this paper, we believe that the combination of SST and ResNet is the optimal signal processing and classification method because they can make full use of the time-frequency characteristics of the signal and have high computational efficiency and classification accuracy.

Figure 8 illustrates the confusion matrix of the original data, showing the classification results and errors of different deep network models with respect to different events. The most confusing events were mainly the events CH, WA, PV, TSLF, and TFLF. These events were characterized by smaller amplitude, lower frequency, or were simply more unstable, so they were not distinctive enough in the time-frequency domain and were easily confused with other events. In order to improve the classification accuracy of these events, this paper suggests using the SST ResNet model, as it can utilize the phase information of the signals to enhance their feature representation and it has high time-frequency resolution and classification performance. As it can be seen in Figure 8, the SST ResNet model attained higher classification accuracies than the other models for all of these events and attained the best overall classification accuracy, i.e., 96.74%.

#### 3.2.2. Attenuation Data Recognition

In order to consider the impact of vibration signal attenuation on the classification performance, the exponential attenuation process was conducted on the actual tap vibration event dataset to simulate the propagation of the signal at different distances and locations, generated by applying the exponential decay shown in Figure 9 to the original signal. Attenuation of vibration signals refers to the phenomenon whereby the amplitude of vibration signals gradually decreases with increasing time or distance. The attenuation of vibration signals may be caused by the dampening of the vibration system, energy loss, absorption, and/or scattering of the propagation medium, all of which will affect the quality and reliability of the signal. The signals of distributed systems are also affected in practice by the monitoring distance, monitoring location, and so on [18]; therefore, the collected data are subjected to exponential attenuation to verify the accuracy of different methods in the case of attenuation. Figure 10 shows the time-frequency plots of CWT, SST, and STFT for the attenuated signal.

From Table 4 and Figure 11, it can be observed that the classification performance of different time-frequency analysis methods and different deep network models on the fading dataset has obvious differences. For the six intensity signals, the SST ResNet model attained the highest recognition accuracy, i.e., 98.70%, and the CWT ResNet model was the second best, with a recognition accuracy of 97.14%. For the recognition accuracy of the six intensity signals under the six background signals, SST ResNet performed best with 95.18% recognition accuracy, followed by CWT ResNet with 94.40% recognition accuracy. These results indicate that the SST ResNet model is able to effectively differentiate between different intensities and types of percussive vibration events and has high robustness and generalization ability even in the presence of signal attenuation.

For the three models, the overall recognition rate and speed are consistent with the characteristics of the original data. This indicates that the performance of different deep network models on the attenuated dataset is consistent with the performance on the original dataset, but there are some differences. With respect to the attenuated vibration data, SST was found to attain the best classification performance under the ResNet model, although the overall recognition rate was still lower than the unattenuated original data. This suggests that signal attenuation has a certain impact on the classification performance and needs to be considered for compensation or adjustment in practical applications.

Figure 12 presents the confusion matrix of the attenuated data, which illustrates the effect of signal attenuation on the classification performance of different models. As shown in the figure, signal attenuation introduces more confusion among the events, especially for those with smaller amplitudes, such as WA and PV. This is because signal attenuation lowers the SNR and amplitude of the signal, which are crucial features for distinguishing different types of events. The proposed model relies on the amplitude and frequency characteristics of the signal to classify the events, but these characteristics may be distorted or submerged by the noise and interference in the field environment. Therefore, the proposed model may not achieve the same high recognition rate as in the laboratory setting. Among the tested models, the SST ResNet model was found to attain the smallest classification error on the attenuated data, followed by the CWT ResNet model. The other models attained larger classification errors, indicating that they are more sensitive to signal attenuation.

## 4. Conclusions

The purpose of this paper is to explore the use of time-frequency diagrams and deep learning models to identify and analyze the vibration signals collected in distributed fiber optic monitoring systems. In this study, different types and intensities of vibration signals were firstly collected in the laboratory, including six kinds of tapping signals with different strengths and six types of vibrations with different events, and then three deep learning models (VGG, ViT, and ResNet) were used to classify and compare the three kinds of time-frequency diagrams (CWT, SST, and STFT), respectively. The results show that the classification accuracy of the Resnet model under the SST time-frequency diagram is optimal, reaching 96.74%. Finally, in order to simulate the signals at different distances, the data were attenuated, and the results show that the classification accuracy remained as high as 95.18%, which indicates that effective identification can be carried out even at a certain distance from the optical fiber. It can be seen that the method proposed in this paper provides a new idea and technical means for distributed fiber monitoring.

## Figures and Tables

**Figure 1 sensors-23-08773-f001:**
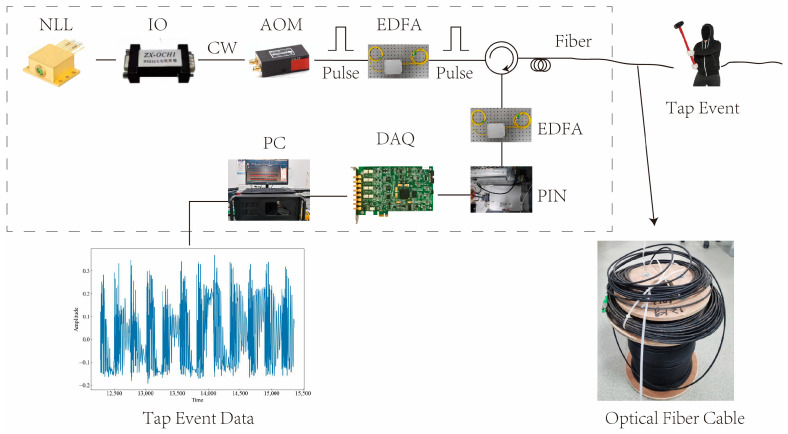
Structure of the Φ-OTDR system.

**Figure 2 sensors-23-08773-f002:**
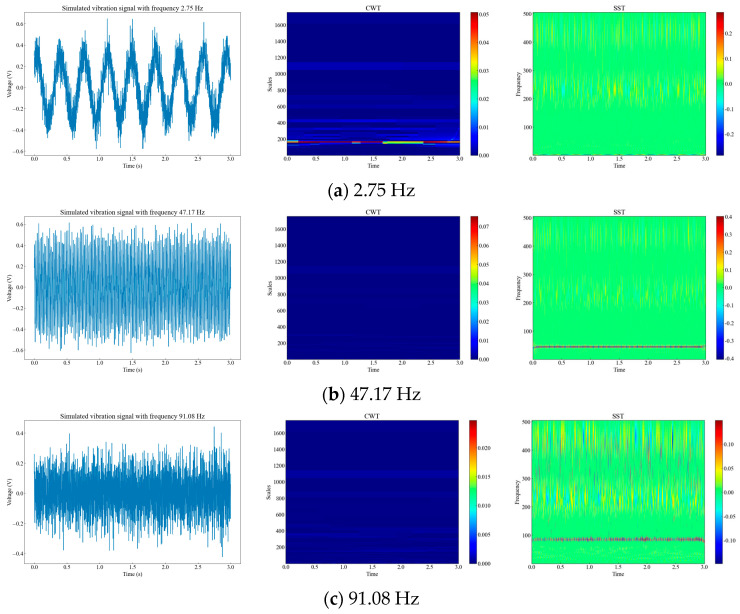
Time-frequency diagrams of three different analog signals. (**a**) Diagrams of 2.75 Hz raw signal, CWT signal, and SST signal; (**b**) diagrams of 47.17 Hz raw signal, CWT signal, and SST signal; (**c**) diagrams of 91.08 Hz raw signal, CWT signal, and SST signal.

**Figure 3 sensors-23-08773-f003:**
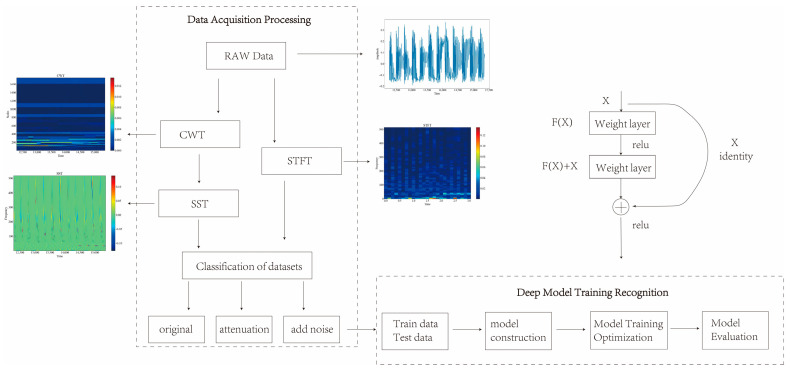
Flow of recognition algorithm.

**Figure 4 sensors-23-08773-f004:**
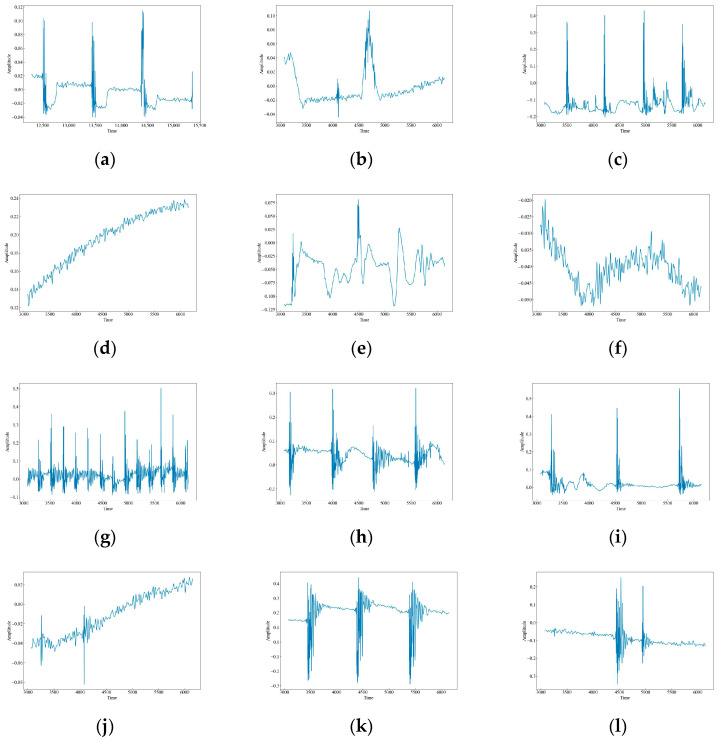
Waveform images of the original vibration signals. (**a**) CH; (**b**) HSZ; (**c**) MI; (**d**) PV; (**e**) ST; (**f**) WA; (**g**) TFF; (**h**) TNF; (**i**) TSF; (**j**) TFLF; (**k**) TNLF; (**l**) TSLF.

**Figure 5 sensors-23-08773-f005:**
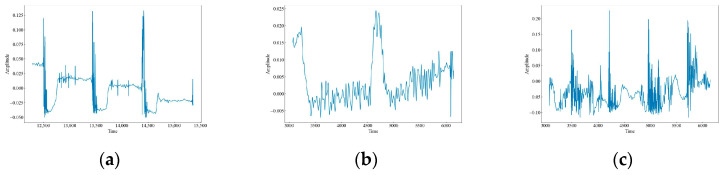
Waveform images of the attenuation vibration signals. (**a**) CH; (**b**) HSZ; (**c**) MI; (**d**) PV; (**e**) ST; (**f**) WA; (**g**) TFF; (**h**) TNF; (**i**) TSF; (**j**) TFLF; (**k**) TNLF; (**l**) TSLF.

**Figure 6 sensors-23-08773-f006:**
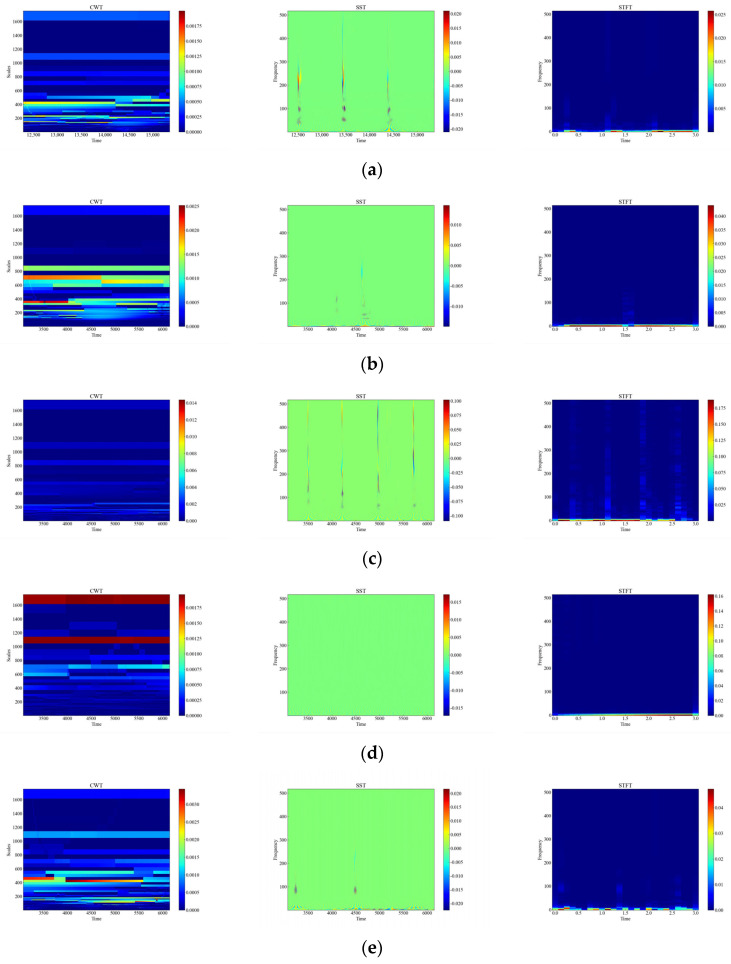
CWT, SST, STFT time-frequency diagram of the original signal. (**a**) CH; (**b**) HSZ; (**c**) MI; (**d**) PV; (**e**) ST; (**f**) TFF; (**g**) TFLF; (**h**) TNF; (**i**) TNLF; (**j**) TSF; (**k**) TSLF; (**l**) WA.

**Figure 7 sensors-23-08773-f007:**
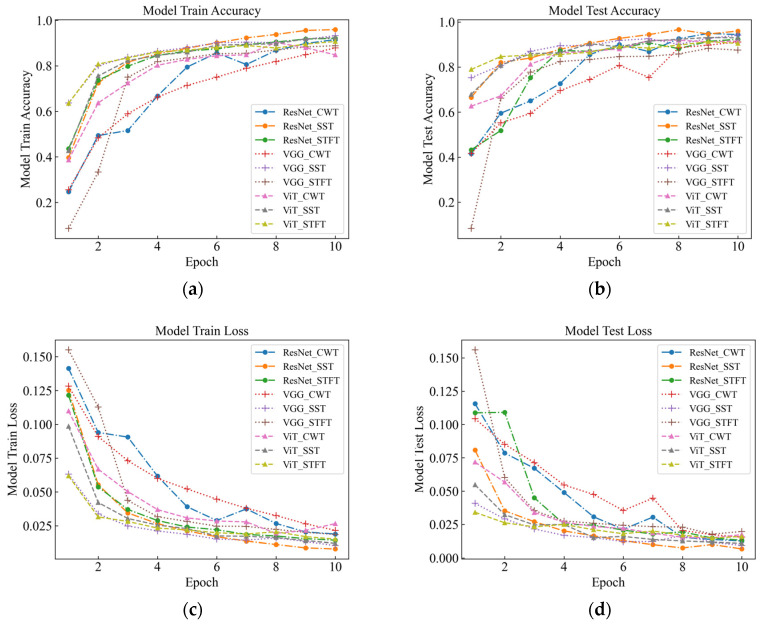
Signal Test classification knot diagram for the original data. (**a**) Model Train Accuracy; (**b**) Model Test Accuracy; (**c**) Model Train Loss; (**d**) Model Test Loss.

**Figure 8 sensors-23-08773-f008:**
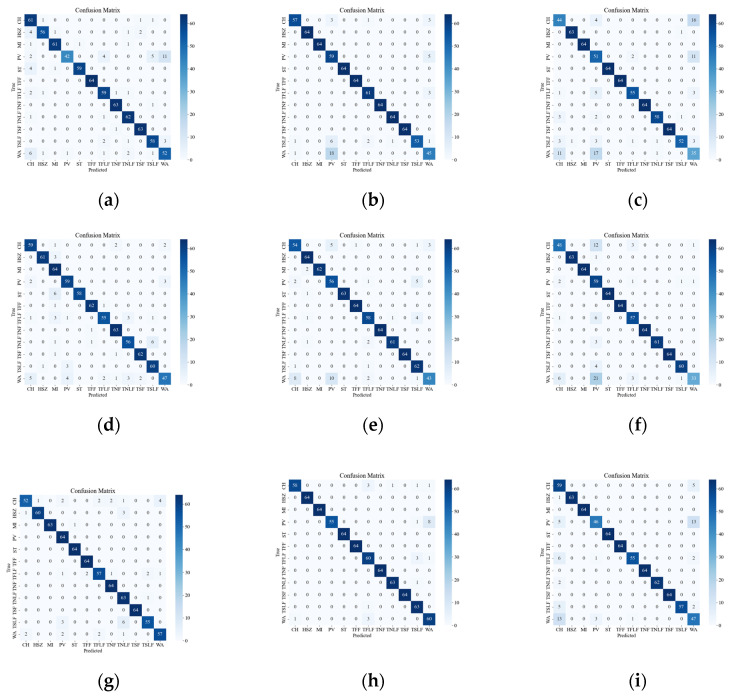
Classification results of the original event confusion matrix. (**a**) VGG CWT; (**b**) VGG SST; (**c**) VGG STFT; (**d**) ViT CWT; (**e**) ViT SST; (**f**) ViT STFT; (**g**) ResNet CWT; (**h**) ResNet SST; (**i**) ResNet STFT.

**Figure 9 sensors-23-08773-f009:**
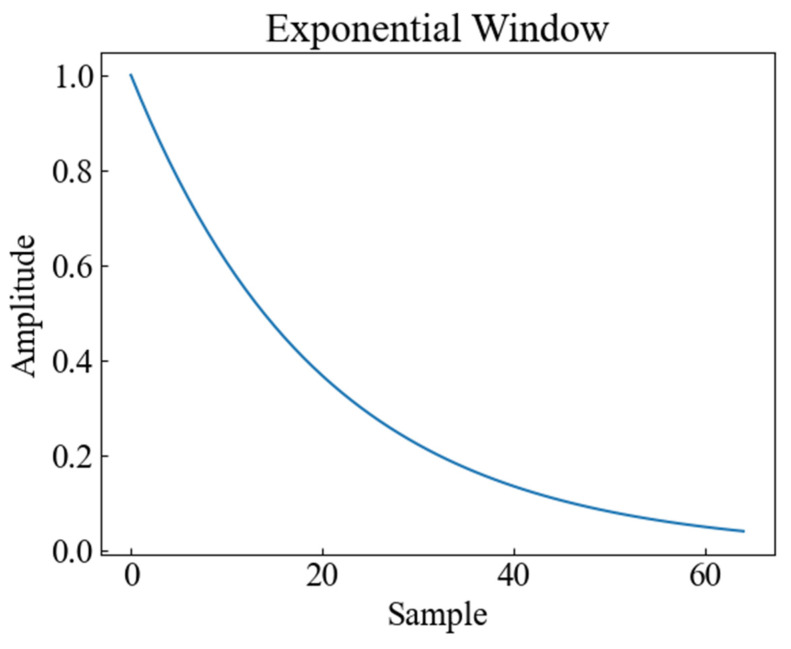
Exponential attenuation signal graph.

**Figure 10 sensors-23-08773-f010:**
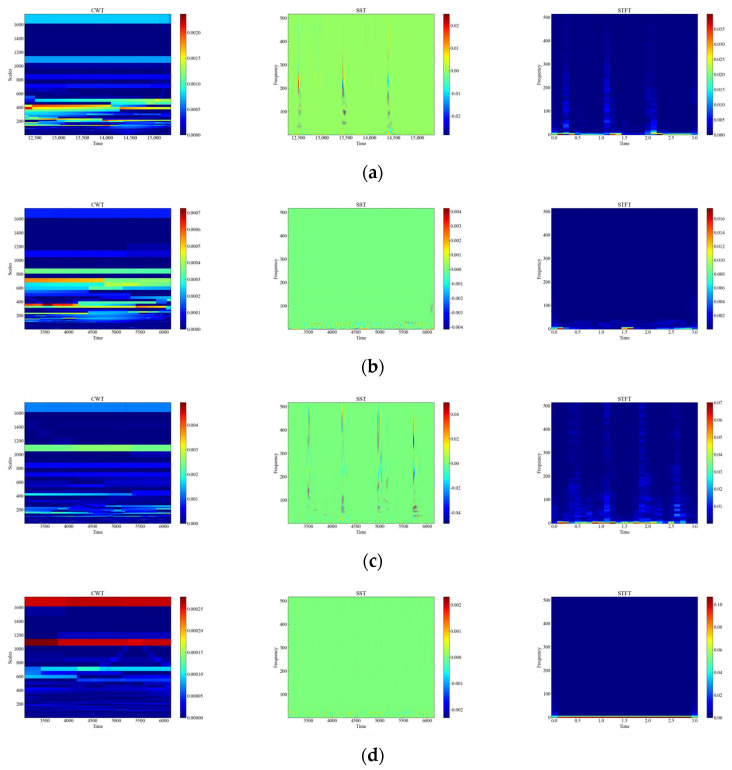
Time-frequency plot of attenuation signals. (**a**) Attenuation of CH; (**b**) Attenuation of HSZ; (**c**) Attenuation of MI; (**d**) Attenuation of PV; (**e**) Attenuation of ST; (**f**) Attenuation of TFF; (**g**) Attenuation of TFLF; (**h**) Attenuation of TNF; (**i**) Attenuation of TNLF; (**j**) Attenuation of TSF; (**k**) Attenuation of TSLF; (**l**) Attenuation of WA.

**Figure 11 sensors-23-08773-f011:**
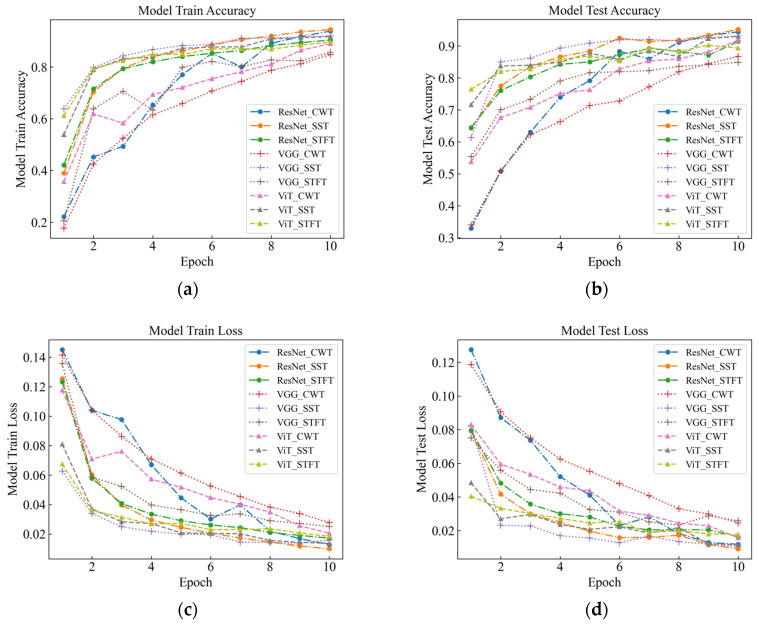
Classification knot diagram for attenuation data testing. (**a**) Model Train Accuracy; (**b**) Model Test Accuracy; (**c**) Model Train Loss; (**d**) Model Test Loss.

**Figure 12 sensors-23-08773-f012:**
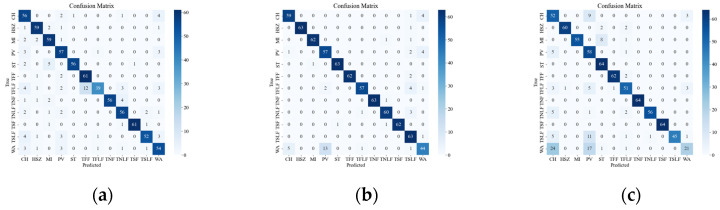
Classification results of attenuation event confusion matrix. (**a**) Attenuation VGG CWT; (**b**) Attenuation VGG SST; (**c**) Attenuation VGG STFT; (**d**) Attenuation ViT CWT; (**e**) Attenuation ViT SST; (**f**) Attenuation ViT STFT; (**g**) Attenuation ResNet CWT; (**h**) Attenuation ResNet SST; (**i**) Attenuation ResNet STFT.

**Table 1 sensors-23-08773-t001:** Distributed fiber optic vibration signal strength dataset.

Event Type	Training Set/Test Set	Label
TFF	256/64	0
TNF	256/64	1
TSF	256/64	2
TFLF	256/64	3
TNLF	256/64	4
TSLF	256/64	5

**Table 2 sensors-23-08773-t002:** Distributed fiber optic vibration signal event dataset.

Event Type	Training Set/Test Set	Label
CH	256/64	6
HSZ	256/64	7
MI	256/64	8
PV	256/64	9
ST	256/64	10
WA	256/64	11

**Table 3 sensors-23-08773-t003:** Comparison of test results.

Classification Method	Six Intensities			All		
	Accuracy%	F1-Score	Time/s	Accuracy%	F1-Score	Time/s
STFT VGG	94.79	0.95	0.28	88.28	0.89	0.29
CWT VGG	96.61	0.97	0.28	91.15	0.91	0.29
SST VGG	97.66	0.98	0.29	94.14	0.94	0.29
STFT ViT	96.88	0.97	0.05	91.28	0.91	0.05
CWT ViT	97.14	0.97	0.05	91.93	0.92	0.05
SST ViT	97.40	0.97	0.05	93.10	0.93	0.05
STFT ResNet	96.61	0.97	0.09	92.32	0.93	0.09
CWT ResNet	98.18	0.98	0.09	94.66	0.95	0.09
SST ResNet	99.48	0.99	0.09	96.74	0.97	0.09

**Table 4 sensors-23-08773-t004:** Comparison of attenuation test results.

Classification Method	Six Intensities			All		
	Accuracy%	F1-Score	Time/s	Accuracy%	F1-Score	Time/s
STFT VGG	93.23	0.93	0.29	84.90	0.85	0.28
CWT VGG	95.57	0.96	0.28	86.72	0.87	0.28
SST VGG	96.35	0.96	0.29	93.10	0.93	0.28
STFT ViT	93.49	0.94	0.05	90.23	0.91	0.05
CWT ViT	94.27	0.94	0.05	91.80	0.92	0.05
SST ViT	96.09	0.96	0.05	92.97	0.93	0.05
STFT ResNet	96.35	0.96	0.09	91.54	0.92	0.09
CWT ResNet	97.14	0.97	0.09	94.40	0.94	0.09
SST ResNet	98.70	0.99	0.09	95.18	0.95	0.09

## Data Availability

Not applicable.

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
