# Peer review of "Vibration Event Recognition Using SST-Based Φ-OTDR System"

_sensors, 2023, doi:10.3390/s23218773_

Round 1
Reviewer 1 Report
The article presents an SST-based method for vibration event analysis and identification of signals acquired through a φ-OTDR.
The topic addressed by the authors is currently of great interest in research due to its potential impact on applications such as geophysics, oceanography, land security and others.
The paper is very well structured and understandable for an expert in the area. The operation principle of the φ-OTDR is very well written and the results are quite interesting. I think that this work is worth to be published in Sensors.
I leave my comments to improve the paper:
1. In introduction, many acronyms are used without giving its meaning. Please revise that, for instance, STFT, CNN, SVM, VGG, ViT, ResNet, BT neural network,…
2. Line 98: “a laboratory system was built in this paper” – please, revise this sentence. Do you mean that a φ-OTDR system was built, and the experimental setup is shown in figure 1?
3. How many meters of optical fibre were used in this experiment?
4. Is the analysis of the data performed in real time or is the data stored and then processed?
5. In line 421, authors state that “the data in the field with the external interference will reduce the recognition rate to a certain extent”. Could you be more specific? What is the main limitation of the proposed model in a real application?
6. Did authors performed experiments by applying two events (or more) at the same time? Is the model prepared for this?
Reviewer 2 Report
The paper “Vibration Event Recognition Using SST-Based Φ-OTDR System” described the setup of a Φ-OTDR (Optical Time Domain Reflectometer) system, and processed the date obtained from the ΦOTDR system with different methods including synchrosqueezed wavelet transform (SST), continuous wavelet transform (CWT) and short-time Fourier transform (STFT) for analyses of six types of taps and six other events. The authors found that recognition outcome from the SST transformed data processed by ResNet (Residual Network) has highest accuracy for both the original and attenuated vibration signals compared with using VGG (Visual Geometry Group) or ViT (Vision Transformer) as deep classifiers. The method proposed in this paper can achieve high recognition rate, which contributes to the advancement of signal analyses of complicated wavelets. I would recommend its publication on Sensors with following revisions.
1. In Section 1, please provide the full names for the acronyms BP (in line 42), CNN (in line 45), RBS (in line 48), YOLO (in line 49), and SVM (in line 54).
2. In Section 2.1, more details should be given to describe the Φ-OTDR system, for example, what are the brands and models of the NLL, EDFA, IO and AOM devices? What is the type of optical fiber used? How are the events applied to the Φ-OTDR system and what is the distance between the event and the data collection center?
3. In line 197, “by applying taps of different strengths at different locations”, how are the tap strength controlled?
4. In Section 2.2, Ψ f(a,b) in equation (1) is CWT of the signal f. The left side of equation (2) is the instantaneous frequency of signal f. In equation (1) and equation (2), their left sides have the same Ψ f(a,b). Please use another form of expression to indicate the instantaneous frequency in equation (2).
5. In Table 1 and Table 2, why the first row appeared at the bottom row once again?
6. For Figure 6 and Figure 9, please label each plot as a, b, c, d and indicate the exact title for each plot.
7. In Section 3.2.2, please indicate in detail about how to obtain the exponential attenuation data. How are the attenuation signal of the events collected? I would suggest adding a figure of the attenuation signal (similar to Figure 4), i.e., the attenuation data used for calculating the CWT, SST and STFT plots in Figure 8.
8. Typos to be corrected: In line 132. “wavelet function .” should be “wavelet function.”; In line 147, “point a?,, and the corresponding T?(?, ?) of …” should be “point a?, the corresponding T?(?, ?) of …”.
